# The Role of Metabolic Engineering Technologies for the Production of Fatty Acids in Yeast

**DOI:** 10.3390/biology10070632

**Published:** 2021-07-08

**Authors:** Numan Ullah, Khuram Shahzad, Mengzhi Wang

**Affiliations:** 1College of Animal Science and Technology, Yangzhou University, 48 Wenhui East Road, Wenhui Road Campus, Yangzhou 225009, China; numanhashmi25@gmail.com; 2Department of Biosciences, COMSATS University Islamabad, Park Road, Islamabad 45550, Pakistan; khuram_shahzad@comsats.edu.pk; 3State Key Laboratory of Sheep Genetic Improvement and Healthy Production, Xinjiang Academy of Agricultural Reclamation Sciences, Shihezi 832000, China

**Keywords:** metabolic engineering technologies, fatty acid production, fatty acid in yeast

## Abstract

**Simple Summary:**

Metabolic engineering involves the sustainable production of high-value products. *E. coli* and yeast, in particular, are used for such processes. Using metabolic engineering, the biosynthetic pathways of these cells are altered to obtain a high production of desired products. Fatty acids (FAs) and their derivatives are products produced using metabolic engineering. However, classical methods used for engineering yeast metabolic pathways for the production of fatty acids and their derivatives face problems such as the low supply of key precursors and product tolerance. This review introduces the different ways FAs are being produced in *E. coli* and yeast and the genetic manipulations for enhanced production of FAs. The review also summarizes the latest techniques (i.e., CRISPR–Cas and synthetic biology) for developing FA-producing yeast cell factories.

**Abstract:**

Metabolic engineering is a cutting-edge field that aims to produce simple, readily available, and inexpensive biomolecules by applying different genetic engineering and molecular biology techniques. Fatty acids (FAs) play an important role in determining the physicochemical properties of membrane lipids and are precursors of biofuels. Microbial production of FAs and FA-derived biofuels has several advantages in terms of sustainability and cost. Conventional yeast *Saccharomyces cerevisiae* is one of the models used for FA synthesis. Several genetic manipulations have been performed to enhance the citrate accumulation and its conversation into acetyl-CoA, a precursor for FA synthesis. Success has been achieved in producing different chemicals, including FAs and their derivatives, through metabolic engineering. However, several hurdles such as slow growth rate, low oleaginicity, and cytotoxicity are still need to be resolved. More robust research needs to be conducted on developing microbes capable of resisting diverse environments, chemicals, and cost-effective feed requirements. Redesigning microbes to produce FAs with cutting-edge synthetic biology and CRISPR techniques can solve these problems. Here, we reviewed the technological progression of metabolic engineering techniques and genetic studies conducted on *S. cerevisiae*, making it suitable as a model organism and a great candidate for the production of biomolecules, especially FAs.

## 1. Introduction

Metabolic engineering is the scientific discipline that is used to study the systematic analysis of biological pathways with molecular biology techniques to modify the metabolic potential and genetics of microorganisms in order to produce desired products. It deals with metabolic fluxes and their reactions to determine the metabolic functions in systems biology [1].

Metabolic engineering took its first leap when genes were introduced into bacteria for the first time in 1973 [2]. Soon after the production of insulin [3], it was assumed that the production of molecules could be possible with the insertion of a single gene into microbes. However, the analysis of ethanol’s overproduction made scientists realize that the process is more complex than previously thought, and it involves manipulating several genes responsible for ethanol production [4]. Thus, attention turned into diversifying techniques for the production of chemical components. A large number of articles were published in the 1990s addressing the science of metabolic engineering [5,6]. The researchers primary goal was to manufacture chemicals, including other applications such as bioremediation and bio-sensing [7]. Overall, metabolic improvement falls into three categories. First, enhancements in the productivity or yield of the desired product are usually achieved by the overexpression of genes that code for a particular protein that helps produce the desired product. Second, byproducts’ production is reduced by removing genes (or genes) that code for a rate-limiting enzyme [8]. The metabolic flow of a pathway can be redirected by the deleting genes involved in the accumulation and production of non-productive reactions. Third, increasing the precursor supply and improving product efflux are performed to increase the yield.

This review discusses the technological advancement for fatty acid (FA) production in yeast and other microbes, ranging from classical techniques such as random mutagenesis to cutting-edge technologies such as CRISPR and synthetic biology. We also discuss different sources of FA production and gene manipulation techniques such as recombinant DNA technology, synthetic biology, and CRISPR–Cas, with a particular focus on *S. cerevisiae*.

## 2. Microbes as Cell Factories in Metabolic Engineering

The sustainable production of biomolecules requires the engineering of microbial cell factories using metabolic engineering principles. They can convert inexpensive feedstocks into chemicals and fuels [9]. The traditional ways to produce these chemicals and fuels have not been able to meet the required demand. Due to economic and ecological goals, microbial production of these chemicals and fuels is growing. However, the efficiency of these microbes to produce these chemicals and fuels is low, which has shifted the focus on engineering microbial-based cells into fully-fledged microbial factories, leading to an alternative approach that has the potential to remove the bottlenecks linked with the other production routes.

Preferably, a microbial cell factory should possess specific characteristics that enable it to produce desired products from the desired feedstock. There are three main features such as (i) enough basic knowledge of the microbe, modeling of media cultures, and bioprocesses, and (ii) the information of dietary requirements. Typically, microbes that can feed on simpler carbon and nitrogen sources are preferred; (iii) resistance to various stresses is crucial since it can affect the overall productivity. Other factors, such as tolerance to harsh conditions, such as high temperature and pH in the fermenter, are also considered [10]. Ideally, the selection of microbial chassis for the biosynthesis of FAs is performed based on their extensive engineering capacity to produce oleochemicals at a high titer [11]. In addition, the engineering strategies in these microorganisms should enhance the supply of metabolic precursors for FAs, such as acetyl-CoA and redox cofactor NADPH.

Moreover, heterogeneous enzymes should also be implemented to functionalize fatty acyl-CoAs into desired derivatives [12]. Currently, model organisms such as *E. coli* and *S. cerevisiae* are mostly being utilized as microbial chassis for the synthesis of FAs and are discussed in detail in the reviews [13,14,15]. However, many other promising organisms are also being explored. For instance, *Rhodococcus opacus*, a chemolithotrophic oleaginous bacterium, has recently been engineered by the deletion of acyl-coenzyme synthetases and the increased exprssion of three lipase enzymes which had lipase-specific foldase to produce 50.2 gL^−1^ of Free fatty acids (FFAs) [16].

Robustness is a major requirement while selecting microbes as a cell factory. However, additional challenges are faced during the production of novel compounds using engineered pathways [17]. Classical studies aiming to solve toxicity rely on physiological or cytological effects that lack understanding of the molecular mechanisms of interaction between the microbe and the toxic compounds [18,19]. However, technological advancements have shifted the focus of toxicological assessment to mechanism-centered analysis at the genome-wide level [20]. More recent techniques to deal with toxicity have relied on altering the membrane’s permeability, a concept often called membrane engineering [21]. For instance, the plasma membrane’s oleic acid can be increased through rat elongase 2 gene overexpression, resulting in the increased tolerance of *S. cerevisiae* to ethanol, propanol, and butanol.

With the development of next-generation DNA assembly and synthesis tools, microbes’ development and characterization as chassis has been in focus. However, not all microbes can be cloned. For instance, more than 99% of the environmental microbes cannot be cultured and studied using the current technologies [22]. Further, foreign genes in a host organism are sometimes expressed weakly or not and are often called “silent” or “cryptic” gene clusters. Other limiting factors include codon optimization between organisms, inadequate precursors and cofactors, and toxicity [23].

## 3. The Importance of FA Production by Microbes

FAs are essential components of cellular processes and are involved in cell signaling and building the phospholipid bilayer of the cell membrane [24]. Generally, FAs are classified into FFAs, fatty alcohols, alka(e)nes, and fatty acid esters [25]. In terms of branching, most natural FAs contain an unbranched chain that ranges from C3 to C28. At the same time, they are classified based on chain lengths into short (≤6), medium (7–12), long (13–20), and very-long-chain FAs (>20) [26]. Industrially, FAs and their derivatives have great importance for producing industrial products such as detergents, soaps, lubricants, cosmetics, and pharmaceutical products [27]. FA-derived components are called oleochemicals. They are also used to produce transportation fuels such as biodiesel [28].

Oleochemicals are primarily derived from plant-based and animal fat sources, which have increasingly concerned sustainability and adverse effects on the environment [29]. Because of environmental sustainability issues such as deforestation associated with the plantations of new oil-based crops, plant sources are not encouraged. However, non-edible oil plants such as jatropha tree (*Jatropha curcas*), karanja (*Pongamia pinnata*), and mahua (*Madhuca indica*) are much more cost-effective as compared to edible oil crops since they can be easily harvested in lands where edible oil crops cannot grow [30]. Some other sources include animal fats, which come as a byproduct of the meat industry and are the cheapest sources. For example, in 2007, the average international prices for poultry fat, yellow grease, and waste cooking oil used for biodiesel production were 256, 374, and about 200 USD/t, respectively, which are 2.5–3.5 times lower as compared to virgin vegetable oils which is 500–800 USD/t [31]. Additionally, to obtain valuable derivatives such as fatty acid alkyl esters (FAAEs)from animal and plant-based fats, the transesterification step is required, making the process expensive.

Recent concerns regarding climate change and crude oil depilation have increased the attention to cleaner biofuels’ production using carbohydrates as a feed source [32,33]. With these concerns growing, alternative sources for cleaner and greener biofuels are being explored. For example, the production of algal biofuels has been a hot topic. Particularly, microalgae strains such as *Chlamydomonas reinhardtii* and *Microchloropsis gaditana*, which can produce lipids up to 60% of its dry weight biomass, have been intensively studied due to their high oleogenic capability [34]. Bioengineering efforts in algae for biofuels have focused mainly on the increased expression of acetyl-CoA carboxylase involved in FA synthesisand blocking starch biosynthesis. Although the overexpression of acetyl-CoA carboxylase in *C. cryptica* and *Navicula saprophila* did not show any increased oil production [35], the overexpression of glucose-6-phosphate in *P. tricornutum* enhanced the production of NADPH, resulting in the increased production of lipids up to 55.7% of its dry weight [36]. Despite successful bioengineering efforts, reviewed elsewhere [37], there are many limitations, and the future of algal fuels is uncertain [38,39]. Algae utilize a lot of water, and larger-scale biofuel production would require water at as large a scale as agriculture. They not only require large surfaces that make it expensive but may face issues such as contamination and maintenance of culture conditions [40]. Moreover, there are certain risks such as harm to the ecosystem structure that may lead to harmful algal blooms and the chance of lateral transfer of genes from algae to other organisms associated with the wide range culturing of genetically modified microalgae [41].

FA production in microbes serves as an excellent alternative for the production of biofulels. Initial studies of the microbial production of FAs took place in the 20th century when evidence of similarities between microbial oil and FAs was found in animals and plants [42]. Moreover, confirmation about the storage capabilities of FAs by microbes was also established around that time [43]. Microbes that could accumulate 20% FAs of the total biomass were termed “oleaginous” [44].

Recent advancements in the field of metabolic engineering have made it possible to industrially synthesize hydrocarbons in microbes. Linear hydrocarbons such as alkanes, alkenes, and esters, particularly hydrocarbons for diesel and jet fuels, can be synthesized by microorganisms using the FA biosynthetic pathway [45,46]. They use low-cost substrates and offer significant advantages in cost and sustainability [47]. Moreover, microbes’ metabolic diversity offers a great range of substrates that can be utilized to produce biofuels. Microbial production of FAs does not need the chemical trans-esterification step required to produce FFAs from vegetables and animal sources.

Another aspect of using microbes is the potential for diversifying the substrate range. This is usually performed by introducing new genes or a pathway enabling the host organism to utilize a new substrate. Recently, Gleizer et al. showed a proof-of-concept study in *Escherichia coli* by utilizing CO_2_ as a substrate to produce all of its biomass [48]. In order to rewire the central carbon metabolism of the *E. coli*, the group co-expressed formate dehydrogenase, phosphoribulokinase, and RubisCO. These three enzymes are involved in CO_2_-fixing during the Calvin–Benson–Bassham cycle. However, the study did not only rely on bioengineering, and adaptive laboratory evolution was also applied to achieved autotrophic growth. This is very significant since the low-cost feed source for producing highly valued products is highly demandable [48]. Similarly, several studies utilizing CO_2_ as an abundant, low-cost feedstock for biochemical production have been conducted [49,50,51]. Altogether, with great potential for FA production and ultimately biofuels, microbes hold immense promise. Yeast, which is generally regarded as safe, is being utilized to produce FA. With current metabolic engineering techniques, they can synthesize FAs and produce enzymes utilized for the industrial production of biomolecules. Recent technological advancements in DNA synthesis, assembly, genome editing, and computational approaches have substantially improved FA production and the production of other compounds in microbes. In *E. coli* and yeast, these advancements have led to the production of artemisinic acid [52], resveratrol [53], FAs [25,54,55,56,57,58], and alkanes [45]. Recently, by engineering synthetic metabolic pathways, the engineered strain of *Rhodococcus opacus* was able to produce higher FAs as well as long-chain hydrocarbons as compared to the wild-type [16]. However, FAs’ microbial production has not reached its full potential as they face problems such as low titer, low yield, slow growth, and low resistance to complex media. Nevertheless, the increased understanding of regulatory networks, the enhancement of gene editing technologies, and new sophisticated combinatorial approaches of metabolic engineering and synthetic biology can overcome the bottlenecks in the production of microbial FAs.

## 4. Genetic Manipulations in FA Production

The majority of the genetic manipulations in microorganisms have targeted *E. coli* to produce FAs. Plant-based, renewable feedstocks, for example biomass-derived carbohydrates, have been utilized to produce FAs in *E. coli*. Steen et al. [25] synthesized fatty esters (biodiesel), fatty alcohols, as well as waxes in engineered *E. coli* by expressing the native *E. coli* thioesterase, tesA (a “leaderless” version of TesA), and also eliminated FadD and FadE, the first two competing enzymes which are involved in b-oxidation. Similarly, about 11% more FAs were produced in an engineered *E. coli* than the control through the over-expression of *fabD* from four different sources. The well-studied and characterized E. coli *fabD* gene encodes malonyl CoA-acyl carrier protein transacylase, which catalyzes malonyl-CoA to malonyl-ACP [59]. An increase in the amount of FFAs was observed in the strains having the *fabD* gene from *E. coli*, *Streptomyces avermitilis* MA-4680, or *Streptomyces coelicolor* A3(2). However, the strain carrying the *fabD* gene from *Clostridium acetobutylicum* ATCC 824 was found to produce the same amount of FFAs as the control strain. Notably, the overexpression of the *fabD* gene was performed by an optimized gene construct. The *fabD* gene was cloned to form an operon with the TE gene, and hence achieving the overexpression of the gene. The study suggested that FFA production improvements can be brought through the overexpression of the *fabD* gene to increase malonyl-ACP activity [60].

Similarly, the level of an omega-hydroxy fatty acid, 3-hydroxy propionic acid production, has shown an increased titer up to 1.8 gL^−1^ h^−1^, 1.4-fold higher than the wild type. In the study, Chu et al. increased FA production by identifying and cloning novel aldehyde dehydrogenase, *GabD4* from *Cupriavidus necator*. Upon investigation, GabD4_E209Q/E269Q showed the highest enzymatic activity. Metabolic enhancement in *E. coli* through fatty acyl-ACP reductase from *Synechococcus elongatus* resulted in improvement of both productivity and yield [61]. The endogenous *E. coli* AdhP plays a vital role in reducing fatty aldehydes to fatty alcohols, and thus, the study established an encouraging synthetic route for industrial microbial synthesis of fatty acid in method *E. coli* [62].

Although most of the work for the microbial production of FAs has been performed using *E. coli*, other organisms have also been utilized. *S. cerevisiae* has also been a subject of many genetic manipulations for metabolic engineered products, and FAs are not an exception. In yeast, the majority of metabolic fluxes are involved in ethanol production during the fermentation process. Therefore, most enhanced FA production strategies are devised to redirect the fluxes to go through the ethanol pathway. To increase microbial FA production, the main target is to increase the amount of acetyl-CoA, malonyl-CoA, and fatty acyl-CoA. Thus, genetic manipulations for the enhanced synthesis of FAs often target the genes involved in these molecules’ production. Since yeast (*S. cerevisiae*) does not have enough cytosolic acetyl-CoA production, improving the supply of acetyl-CoA, several genetic manipulations in *S. cerevisiae* have been conducted. One way to increase the amount of acetyl-CoA production is to inactivate its consumption pathways. The glycolytic flux was redirected towards acetyl-CoA by inactivating the genes for ethanol formation and glycerol production. The inactivation was performed using the widely used marker-based homologous recombination, the loxP–KanMX–loxP method [63]. The inactivation of *ADH1*, *ADH4*, *GPD1*, and *GPD2* genes resulted in the production of 100 mg/L FA-derived advanced biofuel, n-butanol [64].

Similarly, in another study, citrate levels were increased by deleting *IDH1* and *IDH2* genes, which play a role in citrate turnover in the TCA cycle through the marker-based homologous recombination. The strategy yielded a 3–4-fold increase in the citrate levels. The excess of citrate was dealt with by the overexpression of a heterologous ATP-citrate lyase (*ACL*), and hence a significant amount of FAs was recorded in the study [65]. Moreover, Zhou et al. reported 10.4 gL^−1^ of FFAs, a 20% increase compared to the previously recorded amount, in engineered *S. cerevisiae* by optimizing a synthetic citrate lyase pathway. This was done by expressing the ATP citrate lyase and malic enzyme [56]. Alongside the optimization of the two genes, the upregulation of the native mitochondrial citrate transporter and malate dehydrogenase was also performed in the study. Additionally, the malate synthesis pathway was tweaked to increase the production of FAs. For instance, FA production was increased from 460 to 780 mg/L, a 70% increase [66], by downregulating malate synthase, cloning of ATP citrate lyase from *Y. lipolytica*, and eliminating the expression of cytoplasmic glycerol-**3**-phosphate dehydrogenase. Altogether, due to these genetic manipulations, 0.8 g/L of FFAs in shake flasks was recorded.

Malonyl-CoA, another important precursor for the biosynthesis of FAs in *S. cerevisiae*, is catalyzed by acetyl-CoA carboxylase (ACC) from acetyl-CoA. Increasing the supply of malonyl-CoA is vital for increasing FAs in yeast. The increased expression of the *ACC1* gene (encodes ACC) and *FAS* gene yields a higher level of malonyl-CoA. In *S. cerevisiae*, the overexpression of *ACC1*, *FAS1*, and *FAS2* enhanced the titer of FAs [67]. Moreover, heterogeneous expression of genes to obtain an increased supply of malonyl-CoA has also been performed. For instance, FAS expression from Brevibacterium ammoniagenes in *S. cerevisiae* resulted in producing 10,498 μg FAEE/g CDW of fatty acid ethyl ester, an increase of about 6.2-fold compared to the wild type [68]. In another study, a more direct approach to increasing the supply of malonyl-CoA was taken. *AAE13,* a plant malonyl-CoA synthetase gene overexpression in *S. cerevisiae*, led to the increased accumulation of lipid and resveratrol of 1.6-times and 2.4-times, respectively [69]. More recently, dCas9 and malonyl-CoA responsive intracellular biosensors were used to identify novel gene expression fine-tuning set-ups to enhance the levels of acetyl-CoA and malonyl-CoA. A total of 3194 gRNAs were used to target 168 selected genes, followed by screening potential malonyl-CoA overproducers using fluorescence-activated sorting assay [70].

On the other hand, another way to increase the oleaginous microbes’ efficiency by increasing NADPH supply is to reduce the acetyl groups (CH3–CO–) in the growing acyl chain of FAs (–CH2– CH2–). In most oleaginous species, the malic enzyme provides most of the NADPH for FA biosynthesis [71]. Additionally, the pentose phosphate pathway (PPP) provides the additional NADPH required. In order to increase the production of FAs, the enzyme involved has been manipulated. A study by Hao et al. [72] shows the overexpression of the genes for glucose-6-phosphate dehydrogenase (G6PD) and 6-phosphogluconate dehydrogenase (6PGD), the enzymes responsible for increasing the supply of NADPH. Because of the overexpression of the genes, the engineered strain produced 8.1 ± 0.5 g/L FAs, which is an increase of 1.7-times [72].

## 5. Recent Advancements of Technologies in Metabolic Engineering

With the recent advancements in the field of metabolic engineering and other biological tools, it has become easier to manipulate organisms for the desired purposes. The construction of living cells using scalable genetic parts can be performed in the modern age by using various genome editing tools, DNA assembly and chemical DNA synthesis tools to introduce the desired functional properties required in microbes [73]. Beyond classical techniques of random perturbations, many other technologies have been utilized for stain improvements. For instance, transposons have been used in metabolic engineering to improve many species [74,75]. In *S. cerevisiae*, transposon insertion mutagenesis has been utilized for enhanced FA production. In the study, five open reading frames (ORFs), *SNF2*, *IRA2*, *PRE9*, *PHO90*, and *SPT21,* through transposon-based insertion, were found to be associated with increased lipid content [76]. Although these ORFs are not directly related with storage of lipid biosynthesis, however, it has been speculated that they may have a role in carbon fluxes towards the storage of lipids. However, the major problem with transposons is random targeting that can be problematic and hard to predict, making them an unpopular choice. Apart from DNA targeting technologies, transcription regulatory technologies have also been utilized. For instance, RNA interference and antisense RNA approaches have been used for this purpose [77,78]. Using RNAi, Takeno et al. improved the FA content of *M. alpine*. They were able to silence the 12-desaturase gene involved in FA synthesis and obtain a higher amount of FAs. More importantly, the RNAi-based silencing did not compromise the FA composition [79].

On the other hand, targeted nucleases that induce double-stranded breaks are of particular interest. These nucleases can create a double-stranded cut, which is repaired by the cell either through a non-homology end-joining repair (NHEJ) or if a repair fragment is available; the path of homology-directed repair (HDR) is taken, which is helpful for the insertion of genes in the genome (Figure 1a). Transcription activator-like effector nuclease (TALEN) has been studied intensively for its activity and specificity [80,81]. TALEN consists of the fusion of two fused DNA binding domains linked to the N-terminal end of the nonspecific FokI nuclease domain that cuts the DNA, requiring two DNA binding domains [82]. TALEN-based metabolic engineering of *Yarrowia lipolytica* has been applied to produce medium-chain FAs by introducing point mutations identified through computer-aided engineering in the ketoacyl synthase domain of FAS. The domain is directly linked with chain length specificity [83]. Furthermore, innovative gene-editing technology, the clustered regulatory short palindromic repeats (CRISPR), is much more convenient. CRISPR works through a guide RNA (gRNA) that is specific for each target site and Cas (CRISPR-associated) protein [84]. The gene-editing technology is relatively more straightforward, cost-effective, seamless, and can achieve marker-free genome edits. Targeting a new gene only requires the redesigning of gRNA.

Although CRISPR/Cas9 is widely applied in metabolic engineering, other CRISPR proteins are also being utilized. For instance, CRISPR Cas12 (also called Cpf1), which offers certain advantages compared to Cas9, has also been utilized for genome engineering [85]. The CRISPR–Cpf1 system can recognize T-rich PAMs (TTTN) as compared to the purine-rich PAMs (NGG) of Cas9, thus extending the target range, and it can create a double-stranded cut in a staggered manner, which is helpful for precise gene editing [86]. Cas12 has a lesser number of off-target effects [87], and it also has dual endoribonuclease and endonuclease activities, which makes it an alternative choice compared to Cas9 [88].

CRISPR-based editing of microbial genes has been applied in bacteria [89] and yeast [90,91]. Moreover, simultaneous targeting of several genes is possible with CRISPR [92], as reviewed elsewhere [93]. Multiple targeting is of great importance for metabolic engineering since most pathways involve multiple gene networks working in a coordinated manner; targeting a single gene may not improve the desired product. Multiplex targeting with CRISPR can solve the problem as several gRNA targets are chosen, and they can be directed towards the desired target at the same time. In *S. cerevisiae*, many CRISPR-mediated genetic manipulations have been conducted to produce desired bioproducts. For instance, it has been applied for the engineering of marker-free xylose-fermenting yeast [94]. In another study, five genes were simultaneously knocked out in the mevalonate pathway, which resulted in a 41-fold increase in the amount of mevalonate [95]. More specifically, CRISPR-base metabolic improvements of FAs have been applied. For instance, CRISPR-based deletion of FA activation genes by Ferreira et al. resulted in a 40-fold increase in the FFA production in *S. cerevisiae* compared to the wild type, deleting genes resposible for the synthesis of storage lipids, FA oxidation, and the conversion of FFAs to fatty acyl-CoA [96] In the study, *FAA1* and *FAA4*, the acyl-CoA synthetase-encoding genes responsible for the deregulation of fatty acid biosynthesis, were removed as a result, and the modified strain was able to produce 35 mg·gDCW^−1^ of FFAs, about 10-fold more as compared to the control strain. Similarly, *POX1*, which encodes fatty acyl-CoA oxidase, was also deleted in the previously modified strain to prevent FFA and acyl-CoA degradation. The deletion resulted in a 58% increase in the production of FFAs, to 53 mg·gDCW^−1^. Furthermore, the deletion of *PAH1*, *LPP1* and *DPP1* resulted in an increase in FFAs to 102 mg·gDCW^−1^, a 98% increase. Phosphatidic acid (PA) is characterized as an important signaling molecule for the regulation of lipid metabolism, but its dephosphorylation results in the formation of diacylglycerols. High levels of PA are linked with the upregulation of the fatty acid biosynthesis machinery. The deletion of genes *PAH1*, *LPP1*, and *DPP1* involved in the dephosphorylation of PA resulted in a 98% increase in total FFAs, 102 mg·gDCW^−1^ [96]. Similarly, Hyoung et al. applied CRISPR-based engineering in *S. cerevisiae* to disrupt the isocitrate dehydrogenase gene of the TCA cycle and, at the same time, HDR-based insertion of the ATP-citrate lyase gene [97]. CRISPR-based engineering achieved ~a 2 times increase in FA production than the non-CRISPR engineered species reported earlier by Tang et al [65].

Moreover, CRISPR can also be repurposed for gene activation and silencing without inducing a double-stranded break [98]. Catalytically dead Cas9 (dCas9) and dCas12a [99] can be used by regulating the expression of a particular gene without inducing a double-stranded break. For example, a 36% increase in the titer of the medium-chain FAs was achieved via CRISPR-mediated interference by repressing the fermentative pathways [100]. In short, the catalytically inactive Cas proteins are incapable of any cleavage activity and can be fused with other functional domains such as transcriptional activator and repressor domains to achieve gene regulatory functions (Figure 1b). Moreover, the nuclease deficient variants of Cas can also be used for targeting several genes simultaneously [70].

Another CRISPR tool is base editing, which is achieved through the fusion of dCas9 with base editors [101]. Two types of base editors are primarily used: adenine and cytidine base editors. These editors are capable of specific nucleotide conversations. This is considered very valuable since it does not require the DSB of DNA and thus eliminates the requirements of repair fragments [102]. Recent advancement in gene-editing technologies, specifically CRISPR, has shown great potential due to its more straightforward design, efficiency, and specificity. There are still many problems, such as the problem of off-target effects, that need to be solved to explore the full potential of CRISPR in metabolic engineering [103].

Although DNA manipulating techniques have played a significant role in advancing metabolic engineering, many problems such as low titer, yield, and tolerance still await enhancement. Traditional DNA manipulation techniques are tedious, time-consuming, and require several restriction digestions, such as ligation-based cloning. On the other hand, synthetic biology provides more sophisticated de novo synthesis and assembly methods of desired DNA that can be used for the reconstruction of desired pathways for different metabolic engineering applications. Furthermore, metabolite sensors are useful tool for metabolic engineering applications. They can be used both for high-throughput screening of high-producers and also for pathway regulation in response to specific metabolic stimuli. These biosensors sense the signals and give an output, which can be fluorescent molecules or regulatory switches. They can be transcription factor (TF)-based, RNA-based, and enzyme-coupled biosensors. In *S. cerevisiae*, fatty acid biosensors have been reported [104]. The bacterial FadR transcriptional repressors were used for the construction of fatty acid sensors. The same group, bacterial constructed xylose biosensors using XylR repressors, can control protein expression upon the detection of xylose [105]. Similarly, a synthetic sensor for the malonyl-CoA in *S. cerevisiae* has been reported [106]. The biosensor was constructed using a codon-optimized FapR and a synthetic promoter, and was combined with a genomic cDNA library to enhance the synthesis of **3**-hydroxypropionic acid in *S. cerevisiae*.

With the advancements in synthetic biology tools, the applications of metabolic engineering can be extended, and the bottlenecks such as product tolerance and low supply of key precursors can be solved. For instance, synthetic biology has been used to enhance FA-derived biofuels in *E. coli* [107] and improve tolerance in *S. cerevisiae* [108].

## 6. Yeast and Production of FAs Using Metabolic Engineering

Yeast is an important organism that has been used extensively as a model organism for molecular biology studies. It is the first eukaryotic organism whose genome has been sequenced [109]. Compared to molds and algae and tolerance towards harsh conditions in industrial production, its high growth rate contributes to the popularity of oleaginous yeast as an ideal organism for producing FAs and next-generation biofuels [110].

A large breadth of genetic information is available about the yeast genome. Various specific databases are available, containing genetics, proteomics, and interactomics information (Table 1). The information has been handy in considering which genes should be selected as a target for metabolic enhancement. Winzeler et al., in 1999, performed splendid work on a project popularly called The *Saccharomyces* Genome Deletion Project [111]. During the project, 6925 *Saccharomyces cerevisiae* strains were constructed, knocking out almost 96% of the yeast genome. The project provided unique knowledge of over 6000 gene disruption mutants and proved to be of great importance for future metabolic engineers. Apart from metabolic engineering uses, yeast has also become a reliable model organism for studying various fundamental studies such as the cell cycle [112], numerous cancers [113], and viruses [114]. Generally, there two methods of designing yeast strains: one involves the introduction of foreign genes to modify its metabolic pathways that can face regulatory issues, and the other without inducing any DNA edits or only introducing genetic material limited to the genus Saccharomyces, which is often called self-cloning [115]. Since these genetic perturbations may also occur naturally by breeding and self-cloning that’s why in several countries, they are acceptable.

In yeast, FA biosynthesis starts with converting acetyl-CoA to malonyl-CoA through the enzyme acetyl-CoA carboxylase in the cytosol. The precursors, malonyl-CoA, and acetyl-CoA are condensed by FA synthases (FASs) to FAs using malonyl-CoA as the extender unit. Each elongation of two-carbon units in FA biosynthesis requires two NADPH. FAs in the cytosol of *S. cerevisiae* are catalyzed by the type I FAS system and have two functional subunits: the α-subunit and β-subunit encoded by *FAS2* and *FAS1*, respectively [116,117]. Although type II FAS is also responsible for FAs, most FAs in *S. cerevisiae* are synthesized by the type I FAS system. Principally, there are two ways of generating acetyl-CoA in yeast cells. First, CoA is generated from the glycolysis of fermentable sugars through the pyruvate–acetaldehyde–acetate pathway through cytoplasmic acetyl-CoA synthases activity. The second source for acetyl-CoA synthesis in *S. cerevisiae* is the excess of citrate (Figure 2). This excess amount of citrate is transported with the help of citrate transport protein [118]. Citrate is finally catalyzed by ATP-citrate lyase, an enzyme present in all oil-producing microorganisms [119].

The possible way for the metabolic improvement of acetyl-CoA production, as proposed by [120], is first to increase the amount of citrate accumulated in the tricarboxylic acid cycle (TCA cycle), and then transport the excess of citrate in order to generate cytosol acetyl-CoA. This is a promising pathway to provide precursors for FA biosynthesis. Another aspect of the improved production of FAs in *S. cerevisiae* is citrate catabolism. During the TCA cycle, citrate catabolism is performed by an enzyme of mitochondrial origin, isocitrate dehydrogenase. The enzyme is coded by two distinct genes, *IDH1* and *IDH2* [121]. The accumulation of citrate is directly linked to its consumption and hence to the enzyme activity, i.e., if the enzyme activity is high, there is less accumulation of citrate and less chance of producing a high amount of acetyl-CoA. To achieve high production, the enzyme activity needs to be reduced. This is performed by disturbing the genes *IDH1* and *IDH2* separately, and then the effect on citrate accumulation is measured compared to the wild-type strains. The double gene disturbance resultantly yielded a 4 to 5-fold enhancement in citrate level in the media. This suggested that since two genes and those genes code, the enzyme that catalyzes citrate was disturbed in the process, resulting in the net increase in citrate accumulation in the media [65].

Furthermore, the production of a surplus amount of citrate is not only required; instead, the accumulated citrate should also be converted into cytosolic acetyl-CoA. This is performed by the enzyme ATP-citrate lyase (*ACL*). Several studies regarding the effect of *ACL* on the cytoplasmic acetyl-CoA generation or FA accumulation have been conducted [122,123]. These studies established a relationship between *ACL* action and acetyl-CoA precursor synthesis for FA production. However, despite being present in various organisms, including human cells, animal cells, plant cells, and in certain species of yeasts, the enzyme is absent in the *Saccharomycotina*. Therefore, an obvious target for improving acetyl-CoA production and eventually increasing FA synthesis is the insertion of genes that code for the enzymes responsible for catalyzing the accumulated citrate to acetyl-CoA in the cytosol [65]. Similarly, in another study, pyruvate was directed towards endogenous cytosolic acetyl-CoA biosynthesis by deleting *IDH1* and expressing *ACL* from *Aspergillus nidulans* for production mevalonate [124]. This strategy of directing flux through the *ACL* pathway by increasing citrated supply and the expression of active *ACL* can be applied to synthesize acetyl-CoA-derived molecules such as FAs.

## 7. The Expectations in FAs Produced by Metabolic Engineering in Yeast

The production of both small and large molecules has been performed in yeast for many years. These include alcohols, hydrocarbons, and proteins. Yeast, like *E. coli* also holds certain advantages. It grows fast and utilizes cheap carbon sources, and it has very well-established genetics, thanks to the amount of work performed on its genome [125]. Moreover, its robustness and tolerance to the harsh conditions experienced during industrial-scale cultivations have made it an attractive choice for the production of biomolecules.

In recent years, sequencing and assembly technologies have significantly improved, and their cost has dropped exponentially. Moreover, these technological improvements have resulted in a surge of vast amounts of data about biosynthetic genes (BSGs) and pathways responsible for producing natural products [126,127]. A notable example of omics-based BSG discovery is the identification of over 33,000 putative bacterial gene clusters during the analysis of 1154 diverse bacterial genomes [128]. In another study, Xiaoyu Tang et al. discovered a bacterial gene cluster for thiotetronic acid antibiotics, a class of compounds that blocks FA synthesis in *Salinispora* strains [129]. Furthermore, in *P. pastoris*, transcriptomic data were utilized to mine and characterize novel methanol inducible promoters by [130]. In *E. coli*, similar efforts have been made to discover burden-specific promoters using transcriptomic data and to generate a metabolic-burden responsive feedback circuit for controlling the expression of genes [131]. With such an amount of information about gene and metabolic pathways and the introduction of fast-paced and cheaper technologies, manipulating metabolic pathways to produce the desired products has become much more accessible and rapid than ever before.

However, there are still many hurdles, such as poor predictive models due to the inherent non-linearity of biological systems, the lack of widely applicable and scalable part libraries, and low-throughput characterization techniques.

Non-conventional yeasts are also gaining much attention and are being characterized due to their beneficial traits, such as tolerance towards high temperatures and pentose sugar conversion. Although these non-conventional yeasts lack genetic tools compared to *S. cerevisiae*, efforts are being made to use their beneficial traits [132,133]. For instance, non-conventional yeast such as *Kluyveromyces marxianus* has been studied due to its thermotolerance [134] and *K. marxianus* and *Ogataea* for their resistance towards hydrolysate-derived inhibitors [135]. Another non-conventional yeast, *Y. lipolytica*, is of particular importance due to its oleaginous properties and is becoming a competitor of *S. cerevisiae* due to the rapid development of its assembly tools using synthetic biology and CRISPR–Cas [136,137,138,139,140,141]. In *Y. lipolytica* an increase titer of the lipid has been achieved through the inter-convertsion of excess NADH into NADPH [142], reducing glycogen storage [143], and engineering longer-chain FAs into it [144]. Due the recent developments of genetics based tools for non-conventional yeast, it is expected that non-conventional yeasts could be extensively engineered for the synthesis of biomolecules.

As new metabolic engineering, synthetic biology, and genome-editing tools and technologies become available, it will become quicker and cheaper to engineer yeast strains. Altogether, combinatorial approaches utilizing traditional and modern tools (see Figure 3) to produce FAs need to be taken to achieve high titer rates, enhanced production, and improved tolerance in microbes. With new tools and technologies for precise modification of genes and pathways and increased system-level understanding of cellular pathways, the bottlenecks in FA production, such as robustness and low supply of precursors, would be solved. For example, to address the issue of imbalance in the metabolic network faced while achieving an increase in the supply of precursors such as malonyl-CoA through the activation or deactivation of the specific enzymes and metabolic pathways, Qiu and his colleagues constructed a malonyl-CoA repressive biosensor in yeast by the fusion of transcriptional activation domain with *FapR*. The biosensor achieved an 82% repression ratio when the malonyl-CoA was at a high level [145]. Principally, these biosensors sense the signals (malonyl-CoA in this case) and give an output signal that can fluoresce, regulatory switches, or antibiotic resistance. Their application in yeast has been reviewed elsewhere [146]. Furthermore, such a biosensor in yeast for acetyl-CoA has also been developed [147].

Classical metabolic engineering methods were based on random modifications through different mutagens such as UV and certain mutagenic chemicals, followed by screening the modified strains for overproduction. Molecular biology techniques such as recombinant DNA technology rely on restriction enzymes and different cloning methods to enhance strains for the overproduction of a particular molecule. Targeted gene-editing technologies such as different CRISPR–Cas proteins with various functions have made the process of targeted gene editing fast, efficient, and cost-effective. The CRISPR-Cas tool kit has been expanding ever since its discovery. Modifications such as the fusion of Cas proteins with other effector domains and abolishing the activity of the catalytic domain of Cas proteins are also being applied to achieve desired metabolic engineering goals. System and synthetic biology have been used to create genome-scale metabolic models and predict and construct indigenous and novel/non-natural pathways.

## 8. Conclusions

Metabolic engineering has been able to engineer microbes for chemical production and continues to do so. Yeast *S. cerevisiae* has remained an important player in the field; like many others, it has produced diverse chemicals, including FAs, which are the critical starting materials for biodiesel production. This has only been possible due to the tremendous amount of work that has been performed on studying the yeast genome. The introduction of non-native pathways into microorganisms and manipulating them to produce the desired product has shown promising results because of the continuous advancements in gene manipulation and analysis techniques. Precise gene-editing technologies such as CRISPR/Cas9 are proving a handful in this regard. FA production in yeast has also been enhanced through several gene insertions and mutations. However, there are still many problems, such as tolerance to the end product, which at high concentrations is usually toxic, thus affecting the cell growth; solving it is not a straightforward task. The comparative cost of yeast-based FAs with animal and plant sources is another limiting factor. With significant technological advancements in the field and an exponential reduction in the cost of sequencing and other such technologies, the field is hoping to offer much more in terms of chemical synthesis from low-cost feedstock and with improving resistance against fermenter conditions. CRISPR/Cas-based genome engineering in yeast has been applied, and it has been extensively used since its discovery for gene manipulations for metabolic engineering applications. Several CRISPR techniques such as base editing, CRISPR inactivation, and silencing have been applied in yeast. More efficient Cas proteins and engineered variants such as Cas12, Cas13, dCpf1, and dCas9-*Fok1* have broader applications, such as multiplex targeting, transcriptional regulation, and less off-target effects, which prove more valuable for genetic manipulation in yeast for FA production. Moreover, synthetic biology has the potentials to play a significant role in the synthesis of non-native products in yeast utilizing cheap feedstock, and thus in the future, it could solve the problem of global warming associated with fossil fuels and prove cost-effective.

## Figures and Tables

**Figure 1 biology-10-00632-f001:**
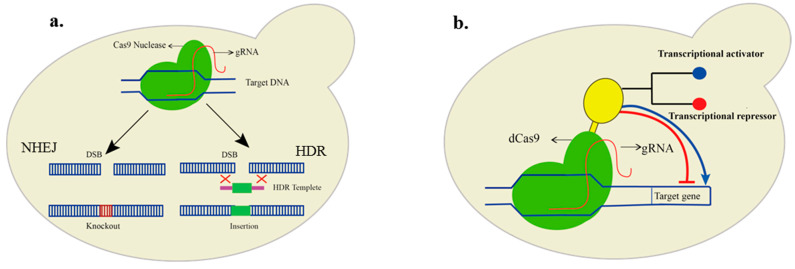
Genetic engineering in yeast using CRISPR/Cas9. (**a**). Cas9, together with guide RNA, induces a double-stranded break (DSB). The DSB is repaired by the cell either through the error-prone non-homology end joining (NHEJ) or in the presence of repair fragments with homology arms; foreign DNA is inserted into the DNA through homology-directed repair (HDR) mechanism. (**b**). The nuclease deficient (dCas9) is fused with a transcriptional regulator (yellow color). In the case of a transcriptional activator (blue), the targeted gene is activated, while in the case of a transcriptional repressor (red), its transcription is deactivated.

**Figure 2 biology-10-00632-f002:**
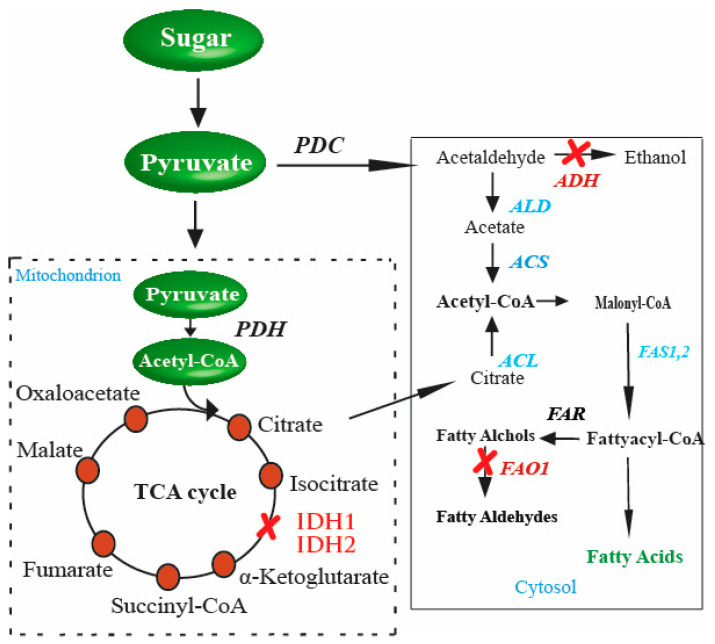
Genetic perturbations in *S. cerevisiae* for enhanced production of FAs. These genetic perturbations result in higher levels of FAs in *S. cerevisiae*. Once there is an excess of citrate in the cytosol, the gene responsible for converting citrate to acetyl-CoA, *ACL* (shown in red in the cytosol), is inserted into *S. cerevisiae*. These genetic perturbations result in higher levels of FAs in yeast. The strengthened steps involved in enhanced FA production are indicated in blue, and the blocked steps are indicated in red color.

**Figure 3 biology-10-00632-f003:**
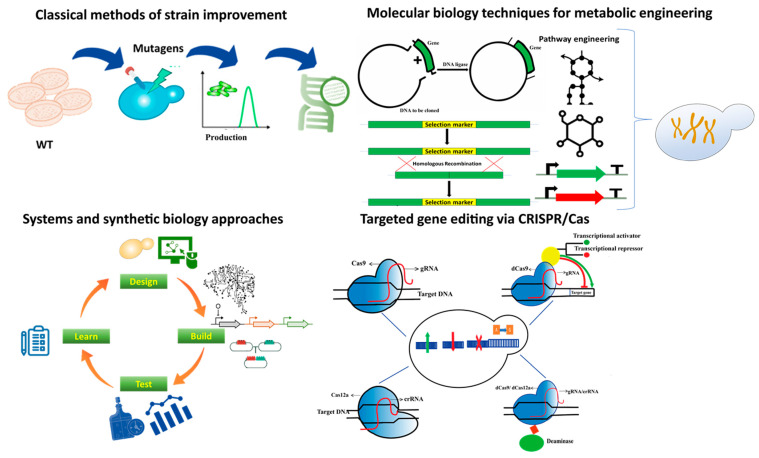
Summary of technologies used in metabolic engineering ranging from classical methods to the state of the art advanced technologies.

**Table 1 biology-10-00632-t001:** Yeast-related online databases and their web links.

Database	Website
Saccharomyces Genome database	http://genome-www.stanford.edu/ (accessed on 25 June 2021)
Yeast deletion project	http://www-sequence.stanford.edu/group/yeast_deletion_project/deletions3.html (accessed on 25 June 2021)
Transcriptional regulatory code of yeast	http://younglab.wi.mit.edu/regulatory_code/ (accessed on 25 June 2021)
Yeast GFP fusion localization database	https://yeastgfp.yeastgenome.org/ (accessed on 25 June 2021)
General repository of interaction datasheets	https://thebiogrid.org/ (accessed on 25 June 2021)
Yeast search for transcriptional regulators and consensus tracking	http://yeastract.com/ (accessed on 25 June 2021)
Cold Spring Harbor laboratory	https://reactome.org/ (accessed on 25 June 2021)

## Data Availability

Not applicable.

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
