# Peer review of "The Role of Metabolic Engineering Technologies for the Production of Fatty Acids in Yeast"

_biology, 2021, doi:10.3390/biology10070632_

Round 1

Reviewer 1 Report

Throughout, the manuscript would still benefit from focusing more on how fatty production is being enhanced by genetically modifying microbes.   

  1. “For instance, Rhodococcus opacus, a chemolithotrophic oleaginous bacterium, has recently been engineered to produce fatty acids (H. M. Kim et al., 2019)” What genes were altered would be very interesting for the reader to know. There is room for specific information here.
  2. “...with the plantations of new oil-based crops.” The authors might address the use of grains such as rapeseed as a source of fatty acid – based lipids without an effect on forests.
  3. “Some other sources include animal fats, which come as a byproduct of the meat industry and are the cheapest sources.” This seems like an off-handed comment. Supplying a value for how much a liter of fatty acids / oil from animals costs to produce vs. a liter of fatty acids / oil from a crop plant costs would support that statement.
  4. “....much progress has been made with the production of fatty acid-derived biofuels in algae, but...” Details about this progress seems central to this review and are worth providing.  Have these algae been bioengineered?  Comparing the pathways of fatty acid production in algae, yeast, and bacteria and how they have been engineered would be interesting.

167 – 173.  Providing more information about the gene(s) involved in bioengineering E. coli to use carbon dioxide as a carbon dioxide as a feedstock would be interesting.

  1. “Similarly, about 11% more FACs were produced in an engineered E. coli than control through the over-expression of fabD from four different sources.” The fact that the fabD protein mediates malonyl- ACP activity is worth explaining earlier in the paragraph.
  2. “Upon investigation, GabD4_E209Q/E269Q showed the highest enzymatic activity.” This is the first mention in the manuscript of using site directed mutagenesis to strategically alter protein structure in line with a bioengineering goal. This is worth describing in some more detail.
  3. “More recently, dCas9 and malonyl-CoA responsive intracellular biosensors...” These biosensors are particularly interesting and should be explained in more detail.

289.” ...five ORFs, SNF2, IRA2, PRE9, PHO90, and SPT21, through transposon-289 based insertion were found to be associated with higher lipid content...” Since those proteins / enzymes are involved in signaling and regulation as opposed to mediating reactions directly in synthetic pathways, it would be interesting to explain how that approach to bioengineering stimulates greater fatty acid synthesis.

  1. “For instance, CRISPR-based deletion of fatty acid activation genes by Ferreira et al., 2018 resulted in 40 fold increase in the free fatty acid (FFAs) production in S. cerevisiae compared to the wild type. Deleting genes involved in the production of storage lipids, fatty acid oxidation, and conversion of FFAs to fatty acyl-CoA.” Since this study is so closely aligned with the topic of this review, it should be described in more detail. What are fatty acid activation genes? The genes involved in the production of storage lipids could be more carefully analyzed. The alteration of which genes / pathways has the greatest promotion of fatty acid synthesis?
  2. Yeast and production of FACs using metabolic engineering” This section has good information flow and depth.

Has there been engineering of bacteria or yeast to influence the length of fatty acids? Has engineering been done to influence then number and position of double bonds? Can these microbes be used to make branched chain fatty acids?

  1. “The introduction of non-native pathways into microorganisms and manipulating them to produce the desired product has shown promising results.” This is an interesting approach to bioengineering microbes as opposed to increasing or decreasing the expression of endogenous genes.  There might be a separate section in the body of the review dedicated to this approach.  
  2. “However, there are still many problems like tolerance to the end product, which at high concentrations is usually toxic, ...” More information about this challenge within the body of the review would enhance the scope and informative nature of the manuscript.

Minor:

Language:

  1. “...such as the low supply of key precursors and the tolerance of product tolerance.”
  2. “Advancements in designing microbial cell factories, specifically yeast, have resulted 18 in producing a plethora of industrially important chemicals.” As stated previously, this is a broad statement that might begin an abstract but it does not belong in the middle of an abstract that is already focusing on fatty acid production.
  3. “(FACS)” I am not familiar with “fatty acids” being abbreviated “FACS”
  4. “Fatty acids (FACS) are essential metabolic requirements ...” An essential metabolite typically means a metabolite that an organism cannot synthesize itself. That is not true for the case of fatty acids in S. cerevisiae.

58.” First, the enhancements in productivity or yield of the desired product 58 are usually done by overexpression of genes that code for a particular product.”  There is room for clarification here.   Using ethanol production as an example, there is no gene that encodes for that particular product. There is a gene that encodes for a protein that facilitates the generation of that product.

  1. “Feed-back inhibition or repression is also vital for increasing productivity, ...” It seems more likely that limiting feed-back inhibiton is vital for increasing productivity.

92.”... metabolic precursors for FACs such acyl-CoAs,”   Acyl-CoA should be replaced by acetyl-CoA since acyl-CoA are commonly made from fatty acids and not the converse.

  1. “Similarly, in another study, citrate levels were increased by the distortion of IDH1 and IDH2 genes.” Distortion is not a commonly used genetic term. Deletion would likely be a better term.
  2. “...optimizing a synthetic citrate lyase pathway through the expression of ATP citrate and malic enzyme...” Presumably, ATP citrate should be ATP citrate lyase.
  3. “A large breath of genetic information...” Please changed breath to breadth.

Author Response

Response to Reviewer 1: We are very thankful to the reviewers about their valuable comments on our manuscript. The critical comments have really improved the manuscript.

“For instance, Rhodococcus opacus, a chemolithotrophic oleaginous bacterium, has recently been engineered to produce fatty acids (H. M. Kim et al., 2019)” What genes were altered would be very interesting for the reader to know. There is room for specific information here.

“...with the plantations of new oil-based crops.” The authors might address the use of grains such as rapeseed as a source of fatty acid – based lipids without an effect on forests.

“Some other sources include animal fats, which come as a byproduct of the meat industry and are the cheapest sources.” This seems like an off-handed comment. Supplying a value for how much a liter of fatty acids / oil from animals costs to produce vs. a liter of fatty acids / oil from a crop plant costs would support that statement.

Response : Thank you for the suggestions. The required information has been added in the manuscript line99, 137-140.  

“                 much progress has been made with the production of fatty acid-derived

biofuels in algae, but.  ” Details about this progress seems central to this review and are worth providing. Have these algae been bioengineered? Comparing the pathways of fatty acid production in algae, yeast, and bacteria and how they have been engineered would be interesting.

Response : Details about bioengineered algae has been added as suggested accordingly. Line 156-170

. Providing more information about the gene(s) involved in bioengineering E. coli to use carbon dioxide as a carbon dioxide as a feedstock would be interesting.

“Similarly, about 11% more FACs were produced in an engineered E. coli than control through the over-expression of fabD from four different sources.” The fact that the fabD protein mediates malonyl- ACP activity is worth explaining earlier in the paragraph.

“More recently, dCas9 and malonyl-CoA responsive intracellular biosensors...” These biosensors are particularly interesting and should be explained in more detail.

Response :The required details has been added to the manuscript  226 , 425, 427

” ...five ORFs, SNF2, IRA2, PRE9, PHO90, and SPT21, through transposon-289 based insertion were found to be associated with higher lipid content...” Since those proteins / enzymes are involved in signaling and regulation as opposed to mediating reactions directly in synthetic pathways, it would be interesting to explain how that approach to bioengineering stimulates greater fatty acid synthesis.

“For instance, CRISPR-based deletion of fatty acid activation genes by Ferreira et al., 2018 resulted in 40 fold increase in the free fatty acid (FFAs) production in S. cerevisiae compared to the wild type. Deleting genes involved in the production of storage lipids, fatty acid oxidation, and conversion of FFAs to fatty acyl-CoA.” Since this study is so closely aligned with the topic of this review, it should be described in more detail. What are fatty acid activation genes? The genes involved in the production of storage lipids could be more carefully analyzed. The alteration of which genes / pathways has the greatest promotion of fatty acid synthesis?

Response: Thank you for pointing out the addition of important details. The details has been added to the manuscript. Line 382-394

Language:

“...such as the low supply of key precursors and the tolerance of product tolerance.”

“Advancements in designing microbial cell factories, specifically yeast, have resulted 18 in producing a plethora of industrially important chemicals.” As stated previously, this is a broad statement that might begin an abstract but it does not belong in the middle of an abstract that is already focusing on fatty acid production.

“(FACS)” I am not familiar with “fatty acids” being abbreviated “FACS”

“Fatty acids (FACS) are essential metabolic requirements ...” An essential metabolite typically means a metabolite that an organism cannot synthesize itself. That is not true for the case of fatty acids in S. cerevisiae.

.” First, the enhancements in productivity or yield of the desired product 58 are usually done by overexpression of genes that code for a particular product.” There is room for clarification here.                                Using ethanol production as an example, there is no gene that encodes for that particular product. There is a gene that encodes for a protein that facilitates the generation of that product.

. “Feed-back inhibition or repression is also vital for increasing productivity, ...” It seems more likely that limiting feed-back inhibiton is vital for increasing productivity.

”... metabolic precursors for FACs such acyl-CoAs,”               Acyl-CoA should be replaced by acetyl-CoA since acyl-CoA are commonly made from fatty acids and not the converse.

“Similarly, in another study, citrate levels were increased by the distortion of IDH1 and IDH2 genes.” Distortion is not a commonly used genetic term. Deletion would likely be a better term.

“...optimizing a synthetic citrate lyase pathway through the expression of ATP citrate and malic enzyme...” Presumably, ATP citrate should be ATP citrate lyase.

“A large breath of genetic information...” Please changed breath to breadth.

Response:    We are thankful to the reviewer for pointing out the the valuable language mistakes . They  have been correcte.

Reviewer 2 Report

Thank you for addressing the majority of concerns. The authors successfully applied the comments in their revisions of the manuscript. As such, it reads much better, and by removing most of the unneeded text allows for more focus on the important sections of the review.

Author Response

We are very thankful to the reviewers about their valuable comments on our manuscript. The critical comments have really improved the manuscript.

Reviewer 3 Report

Ullah et al reviewed the metabolic engineering for the production for the production of fatty acid in yeast. The authors focused the advances on the production of fatty acids by metabolic engineering. The general composition of subparts is well planned, but the content is a bit redundant and difficult to read, some information is misleading. I suggest some points to improve the manuscript as below.

  1. The authors use ‘yeast’ term mostly for Saccharomyces cerevisiae, and this is a bit confused for several sentences. Most of the details about S. cerevisiae only, not for yeast in general, so please use S. cerevisiae specifically.

  1. Generally the abbreviation of fatty acid(s) is FA(s), free fatty acid(s) is FFA(s).

There are many sentences using fatty acids not FAs. Please check all the abbreviations used, define once and then use the abbreviation only.

  1. The manuscript aims to review the production of FA (lipid) from yeast, if it is general yeast not only S. cerevisiae, the oleaginous yeasts which showed a great improvement on lipid production must be explained (general species, lipid synthesis, metabolic engineering for lipid production). In the manuscript, the authors mentioned the oleaginous very briefly which is disappointing.

  1. The authors mentioned the advantage of ME is to FA derivatives not only FA, but the examples or references on FA derivatives are missing.
  2. Line 174, “Microbes like E. coli and yeast, which are generally regarded as safe, “, E. coli is not a GRAS m/o.
  3. Genetic manipulation section, please add table summarized the engineered pathway/lipid production or figure of metabolic pathway which can help to follow the manuscript.

  1. Genetic manipulation section, you need to always provide the gene name, origin, enzyme name. Some sentences are missing the whole information.

  1. When it explains about FA production, one of exact values with the correct dimension among titer, lipid content, productivity, yield should be written not only with improvement (eg. by 4-fold, by 10% increase)

  1. Table 2, please re-verify the websites because some are wrong, and I cannot find some online.

  1. Recent advances in ME tools section, the authors mentioned several tools in the beginning but explained the details of CRISPR only, which is also too long. Please introduce other tools as well as reduce the CRISPR part.

  1. In figure 2, the cell membrane needs to be drawn. If this figure is for S. cerevisiae, please modify the figure legend and indicate the heterologous / native genes.

Minor points

  1. Please check the species names are correctly written (capital at the first, italic, short name after the first use)
  2. Please check the spelling/the space/etc.

Author Response

Reviewer 3

Generally, the abbreviation of fatty acid(s) is FA(s), free fatty acid(s) is FFA(s).

The authors use ‘yeast’ term mostly for Saccharomyces cerevisiae, and this is a bit confused for several sentences. Most of the details about S. cerevisiae only, not for yeast in general, so please use S. cerevisiae specifically.

There are many sentences using fatty acids not FAs. Please check all the abbreviations                                          used, define once and then use the abbreviation only.                

Response : Thank you for the well thoughtful comments. We really appreciated you for taking your time and reviewing our manuscript. The abbreviations have been corrected in the manuscript and in most places where required, yeast has been replaced with S. cerevisiae.

  1. The authors mentioned the advantage of ME is to FA derivatives not only FA, but the examples or references on FA derivatives are
  2. Line 174, “Microbes like E. coli and yeast, which are generally regarded as safe, “,
  3. coli is not a GRAS m/o.
  4. Genetic manipulation section, please add table summarized the engineered pathway/lipid production or figure of metabolic pathway which can help to follow the

Response: Thank you so much for the suggestions. We think genetic manipulation section has a lot of details and figure 2 is enough to summarized the engineered strategies. 

  1. Genetic manipulation section, you need to always provide the gene name, origin, enzyme name. Some sentences are missing the whole

  1. When it explains about FA production, one of exact values with the correct dimension among titer, lipid content, productivity, yield should be written not only with improvement (eg. by 4-fold, by 10% increase)

9.

Response : Thank you for the comments. The websites have been verified, exact values of improvements in lipid content where available, has been added in to manuscript.

  1. Recent advances in ME tools section, the authors mentioned several tools in the beginning but explained the details of CRISPR only, which is also too long. Please introduce other tools as well as reduce the CRISPR
  2. In figure 2, the cell membrane needs to be drawn. If this figure is for S. cerevisiae, please modify the figure legend and indicate the heterologous / native

Response : Thank you for mentioning. Details about other tools have been mentioned including that of non-traditional yeast 335-342 and 425-437. Figure 2 has also been modified accordingly.

Minor points

  1. Please check the species names are correctly written (capital at the first, italic, short name after the first use)
  2. Please check the spelling/the space/etc.

Round 2

Reviewer 3 Report

The authors progressed the manuscript regarding the clarity of terms.

There are still many type errors on gene/species name, please verify thoroughly. 

In figure 2, the newly added cytosol boundary is a bit misleading since glucose and pyruvate are not included in any of the locations. I suggest adding a cell membrane. 

This manuscript is a resubmission of an earlier submission. The following is a list of the peer review reports and author responses from that submission.

Round 1

Reviewer 1 Report

Here in the review article entitled “The Role of Metabolic Engineering Technologies for the Production of Fatty Acids in Yeast …” Ullah et. al, have reviewed the technological advancements in the fields of metabolic engineering and discussed how these are being used in fatty acid production in yeast cells. They have discussed various genetic and molecular technique which are being used in the metabolic engineering field, however, I am sorry if I missed the point, but I think there should be detailed discussion about-

  • Studies which are using CRISPR and other genome editing techniques for the metabolic engineering and fatty acid production.
  • How they are managing the off-target effects and regulating the overall metabolic pathways.
  • How these advance techniques are making the overall process efficient and cost-effective and also is the process scalable.

Author Response

Thank you for reviewing the paper. We highly appreciated you for taking out time and reviewing our paper.

Reviewer 2 Report

The authors have written a manuscript to review developments in genetic manipulation technologies as they apply to fatty acid production in yeast (e.g. S. cerevisiae).  Sections focus on topics such as microbial species, methods of genomic manipulation, and cellular metabolism that generates the precursors for the fatty acid synthase reaction. 

Concerns:

1)   The review would benefit from a more detailed approach.  The extent of the contribution of more modern techniques such CRISPR / Cas to optimize fatty acid production should be related by providing many more specific examples.  In many places, there are broad statements, example of which are below, which would be best supported by following sentences that include detailed methodology and results.  Such sentences may serve well as subject sentences at the beginning of a paragraph.  However, these sentences do not work well when in the middle of a paragraph with no supporting details.  Line numbers are shown.

41:  “Several research articles were published in the 1990’s addressing the science of 41 metabolic engineering.”  

Using PubMed, I found 920 articles published from 1990 – 1999 with the keywords “metabolic engineering”.  That does well beyond “several”. Providing specific examples would give the reader a better sense for how successful metabolic engineering was in the 1990s.  It also bears stating what methods were used since the abstract seems to indicate that cutting edge techniques have provided advanced solutions that previous techniques did not.

187: “Recent technological advancements in DNA synthesis, assembly, genome editing, and computational approaches have substantially improved fatty acid production and other compounds production in microbes.” 

This should be followed by detailed examples.

227 -232:  “The glycolytic flux was redirected towards acetyl-CoA by inactivating the genes ...”  Since this paper has an emphasis on technology, it be related how the genes were inactivated.   Was this via replacement with a marker gene by homologous recombination?  Similarly, it bears stating the technique used to distort genes as indicated in the sentence, “Similarly, in another study, citrate levels were increased by the distortion of IDH1 231 and IDH2 genes ...”

409: “Yeast holds certain advantages over other such organisms that are being used for the production of these molecules. It grows fast and utilizes cheap carbon sources; ..”  Since E. coli is one of the other species mentioned, how yeast grow faster and utilize cheaper carbon sources bears explaining.

416 – 418: “Moreover, these technological improvements have resulted in a surge of vast amounts of data about biosynthetic genes and pathways responsible for producing natural products.”  This should be followed with specific and detail information.

2)  The review would also benefit from going deeper into the pathways that may be manipulated to increase fatty acid production.  Fatty synthase is never mentioned, nor are the enzymes that esterify fatty acids and those that incorporate acyl chains into triglyceride.   Also, the heterogeneity in naturally occurring fatty acids in terms of length and number and position of double bonds is not addressed.  How yeast may be engineered to generate non-native fatty acids with respect to branching (as can be found in plants) could also be addressed.  A new section might be dedicated to the array of fatty acid species that have a commercial use and how each is currently produced by engineered microbes.  A table might be included to summarize such information.  Including a structural diagram of a model fatty acid with indications of where the addition or replacement of functional groups may occur might help.  Some statements that provide examples of this issue are below.

131: “Fatty acids are essential to the cells as they act as an energy substrate.”  Since yeast can live without using the oxidation of fatty acids to yield NADH and eventually ATP, fatty are not exactly “essential” for energy.  They are certainly essential for synthesizing membrane components.

204:  “The study suggested that free fatty acid production improvements can be brought through genetic manipulations to increase malonyl-ACP activity.” It is not clear from the previous sentences which manipulations increased malonyl-ACP activity.   

215:  “...the study established an encouraging synthetic method for industrial microbial production of fatty alcohols in E. coli.” This paragraph began by addressing omega hydroxy fatty acids.  It bears stating whether a hydroxyl on the methyl end of a fatty acid makes it a fatty alcohol or whether being a fatty alcohol requires replacing the carboxylic acid with an alcohol group.

220:  “In yeast, the majority of metabolic fluxes go through ethanol formation during fermentation.   Therefore, most engineering strategies are devised to redirect the fluxes to go through the ethanol pathway.” Is that engineering devised to maximize fatty acid production?

328-329: “With the advancements of synthetic biology tools (Table 1), the applications of metabolic engineering can be extended, and the bottlenecks in the field can be solved.” Where are the bottlenecks in the bioengineering of fatty acid synthesis? That would be a very interesting question to focus upon.

376  - 377.  The appearance of citrate in the media suggests that excess citrate may be exported from the cell as opposed to being acted upon by citrate lyase.  There seems to be room for improvement via metabolic engineering there. 

388 – 391;  “An obvious target for improving acetyl-CoA production and eventually increasing fatty acid synthesis is the insertion of genes that codes for the enzymes that catalyze the accumulated citrate to acetyl-CoA in the cytosol (Tang et al. 2013a).”  It would be appropriate to cite papers such as Rodriguez, S. (2016) “ATP citrate lyase mediated cytosolic acetyl-CoA biosynthesis increases mevalonate production in Saccharomyces cerevisiae” here.   The authors might also address how cytosolic acetyl-CoA can be used in the mevalonate pathway to make isoprenoids and potentially sterols.

431 – 440:  This is the only section that addresses yeast which are naturally oleaginous.  Since this paper is focused on fatty acid production, such species might be addressed in greater depth.

3) The review lacks specific information about how more modern genome manipulation techniques will improve fatty acid production in engineered yeast.  While some details were related about the CRISPR / Cas9 and Cas12 systems, how these have been used to manipulate fatty acid production bears describing.  Aside from speed, how reducing or eliminating specific gene expression by CRISPR – mediated techniques is better than using targeted gene replacement with marker genes bears explaining.  Are the marker genes providing confounding factors? Does CRISPR provide the benefit of providing less than full elimination of gene expression and thus allow a more careful or measured expression reduction?  Also, the manuscript includes  examples of S. cerevisiae genes being overexpressed in S. cerevisiae as well as examples of genes from other species being used.  The strategy for choosing which species’ genes to overexpress is worth relating.  Some examples of statements that bear clarifying are below.

187: “Recent technological advancements in DNA synthesis, assembly, genome editing, and computational approaches have substantially improved fatty acid production and other compounds production in microbes.” 

This should be followed by detailed examples.

262: “For instance, transposons have been used in metabolic engineering to improve many species .”

Are there examples of using transposons to optimize fatty acid production? The same goes for RNA interference. (line 266 – 267)

Figure 1a.  There only seems to be one double stranded break induced.  How NHEJ results in a gene being “knocked out” (which suggests the gene is absolutely dysfunctional) as opposed to NHEJ resulting in a small scale deletion occurring within the gene bears explaining.

Author Response

Thank you for such valuable, technical and detailed suggestions. We highly appreciate your time in reviewing our paper. 

Reviewer 3 Report

Overall, I would say the manuscript is very wordy in places where it is not necessary and lacking in areas that would be more impactful. There are many sections with repetitive or non-informative information that distract from the overall flow of the article. For example, lines 60 through 90 have no substantial impact on the paper (as far as the focus of fatty acid production) and could be removed. Additionally, the same applies to lines 95-113. This section does not add to the paper and could be more impactful if this text were used to give specific examples and breakthroughs in the field of converting microbes to cell-factories. The same can be said about lines 114-121, this is a very important aspect of metabolic engineering and no specific examples are discussed. I do really like the premise of the paper, but it lacks impact as there is limited focus on the most recent advancements in the field and too much time spent discussing advancements like CRISPR and genome sequencing in a broad sense, not how it specifically has been applied to the production of fatty acid in microbes.

Figure 1 is a perfect example of this, you demonstrate the use of CRISPR but fail to link this to fatty acid production in microbes. Instead, this figure could be used to document/demonstrate specific strategies implemented for increasing fatty acid production that have used CRISPR. Just remember, the review is on fatty acid production in microbes and the technology used to reach these goals, this should be the total focus of the entire paper and each section should add to this theme. If it does not have a direct impact on portraying that message, consider just removing it from the manuscript. Table 1 in its present form is quite lacking as a main image for this review. Consider putting this information into a figure that depicts a timeline of events from earliest to most recent, and how these breakthroughs impacted fatty acid production in microbes. The current format is not exciting to look at, so just think of the best way to get this same content into a more impactful format. Table 2 is alright in its present set up, just make sure align everything to make it look cleaner, keep it left-justified and it should be fine. Figure 2 is a little better but is still quite scarce of impact. Keep the biochemical pathway set up used currently, just add a few more strategies of metabolic engineering into the diagram to add to the overall depth of the figure. This is the same for Figure 3, with a large amount of space being used for plain text. Also, this does not depict progression, but an overview of the technologies used, progression implies a linear timeline so I would avoid using a circle. Probably best to just remove the word progression from the figure and make it a summary of technologies. Consider using an image to depict each technology instead of simply listing them. This figure needs a major rework, especially if it will be such a large part of the page and act as a summary for paper.

In summary, make sure to streamline the text to remove unneeded sections that do not directly improve the message, think about the order of sections throughout (take us on a journey, starting with older techniques and breakthroughs ending with the most recent advancements), and make sure that every section has a direct impact on the overall message. Also, spend more time discussing/reviewing more recent advances, as people reading this will most likely have a good understanding of basic molecular biology techniques, and craft your figures to highlight these more recent breakthroughs.

Author Response

Dear reviewer, Thank you for such thoughtful suggestions. Thank you for suggesting changes in the figure for making them impactful.  
